# The utility of intraoperative ultrasonography for spinal cord surgery

Hangeul Park[1], Jun-Hoe Kim[1], Chang-Hyun Lee[1,2], Sum Kim[3], Young-Rak Kim[4], Kyung-Tae Kim[5], Ji-hoon Kim[6], John M. Rhee[7], Woo-Young Jo[8], Hyongmin Oh[8,9], Hee-Pyoung Park[8,9], Chi Heon Kim[1,2,10]*

1 Department of Neurosurgery, Seoul National University Hospital, Seoul, Republic of Korea, 2 Department of Neurosurgery, Seoul National University College of Medicine, Seoul, Republic of Korea, 3 Department of Neurosurgery, Kandong Sacred Heart Hospital, Hallym University College of Medicine, Seoul, Republic of Korea, 4 Department of Neurosurgery, Armed Forces Yangju Hospital, Yangu, Republic of Korea, 5 Department of Neurosurgery, School of Medicine, Kyungpook National University Chilgok Hospital, Kyungpook National University, Daegu, Republic of Korea, 6 Department of Radiology, Seoul National University Hospital, Seoul National University College of Medicine, Seoul, Republic of Korea, 7 Department of Orthopaedic Surgery, Emory University School of Medicine, Atlanta, Georgia, United States of America, 8 Department of Anesthesiology and Pain Medicine, Seoul National University Hospital, Seoul, Republic of Korea, 9 Department of Anesthesiology and Pain Medicine, Seoul National University College of Medicine, Seoul, Republic of Korea, 10 Department of Medical Device Development, Seoul National University College of Medicine, Seoul, Republic of Korea

* chiheon1@snu.ac.kr

## Abstract

### Objectives

Intraoperative ultrasonography (IOUS) offers the advantage of providing real-time imaging features, yet it is not generally used. This study aims to discuss the benefits of utilizing IOUS in spinal cord surgery and review related literature.

### Materials and methods

Patients who underwent spinal cord surgery utilizing IOUS at a single institution were retrospectively collected and analyzed to evaluate the benefits derived from the use of IOUS.

### Results

A total of 43 consecutive patients were analyzed. Schwannoma was the most common tumor (35%), followed by cavernous angioma (23%) and ependymoma (16%). IOUS confirmed tumor extent and location before dura opening in 42 patients (97.7%). It was particularly helpful for myelotomy in deep-seated intramedullary lesions to minimize neural injury in 13 patients (31.0% of 42 patients). IOUS also detected residual or hidden lesions in 3 patients (7.0%) and verified the absence of hematoma post-tumor removal in 23 patients (53.5%). In 3 patients (7.0%), confirming no intradural lesions after removing extradural tumors avoided additional dural incisions. IOUS identified surrounding blood vessels and detected dural defects in one patient (2.3%) respectively.

**Data Availability Statement:** All relevant data are within the manuscript and its Supporting information files.

**Funding:** This study was supported by Ministry of National Defence of Republic of Korea (800-

20230466) and Doosan Yonkang foundation (800-20210527). This study was also supported by grant (30-2023-0120) from the Seoul National University Hospital research fund. The funders had no role in study design, data collection and analysis, decision to publish, or preparation of the manuscript.

**Competing interests:** All the authors confirmed no conflict of interest regarding this study.

## Conclusions

The IOUS can be a valuable tool for spinal cord surgery in identifying the exact location of the pathologic lesions, confirming the completeness of surgery, and minimizing the risk of neural and vascular injury in a real-time fashion.

## Introduction

Ultrasonography is widely used across various aspects of modern medicine, and recently, its utilization in neurosurgery has also increased significantly. In brain diseases, ultrasonography is being utilized for the evaluation of traumatic brain injury, subarachnoid hemorrhage, hydrocephalus, brain tumors, and functional diseases [1–4]. In spinal diseases, ultrasonography is used for the evaluation of dysraphism, spinal tumors, spinal trauma, cerebral palsy, vascular abnormalities, and degenerative diseases [5–9]. Intraoperative ultrasonography (IOUS) is a valuable intraoperative imaging tool as it is portable, provides repeated real-time imaging features, has no radiation exposure, requires no special preparation for patients, and is relatively cheap [10]. In particular, IOUS shows the real-time spinal tumor's extent and location, thereby facilitating the maximum removal of the tumor while minimizing damage to neural structures [11]. Despite the ease and relative affordability of IOUS, and its ability to provide real-time imaging features, its utilization in spinal cord surgery is not yet widespread. This study aims to discuss the advantages of utilizing IOUS in various spinal cord diseases and review related literature.

## Materials and methods

### Patients

All patients who underwent spinal cord surgery using IOUS at a single institution from November 2021 to December 2023 were retrospectively reviewed. All retrospective data reviews were conducted in January 2024. Since November 2021, IOUS has been used in spinal cord surgery when the ultrasonography machine is available. Electric medical records were collected on age, sex, diagnosis, tumor location, preoperative patient-reported functional status (EuroQol 5-Dimension 5-Level; EQ-5D-5L) [12], preoperative McCormick scale [13], operation time, degree of resection (DOR), attenuation of motor-evoked potential, postoperative complications, length of hospital stay, follow-up duration, last follow-up tumor status, last follow-up EQ-5D-5L, and last follow-up MMC. The attenuation of MEP was assessed based on an all-or-none criterion [14]. The DOR was based on preoperative contrast-enhanced T1-weighted and T2-weighted magnetic resonance (MR) images, as well as the first postoperative contrast-enhanced T1-weighted and T2-weighted MR images [15–17]. The pre- and postoperative MR images were interpreted by a radiology specialist to assess the DOR of the tumor. In cases where clinical correlation is necessary in addition to MR imaging for the assessment of DOR, surgical findings were consulted to evaluate the DOR [18, 19]. Surgical records were reviewed to collect information on the timing of IOUS use and the advantages gained from its use.

### Intraoperative ultrasonography

All surgeries were performed in the prone position under general anesthesia. The extent of the surgical approach was determined by referring to preoperative MR imaging and intraoperative

fluoroscopy to check the surgical level. A Laminectomy was performed based on the aforementioned information. After laminectomy and exposure of dura, the location and extent of the tumor were assessed using IOUS. The surgical cavity was filled with sterile saline to serve as a medium for ultrasonographic imaging acquisition. Using a medium, it is possible to acquire ultrasonographic images of the spinal cord disease without direct spinal cord manipulation. The ultrasonography (GE Logiq P9, GE HealthCare, Chicago, Illinois, USA) with linear probe (GE L8-18i-RS Probe; footprint 11.1 x 34.8 mm; bandwidth 6.7–18.0 MHz imaging frequency) was used for spinal cord disease. The probe was coated with ultrasonography-specific transducing gel and then covered with a sterile vinyl surgical sheath. Standard B mode was primarily used to check both axial and sagittal planes. Color Doppler mode was used to assess vessels within or around the lesion. IOUS was primarily used before the opening of the dura, before myelotomy, and finally after the removal of the tumor, as well as other times as needed [20]. The features of IOUS were interpreted by the surgeon in real-time. When the longitudinal window of laminectomy did not encompass the total extent of the tumor, or the axial window was too narrow to remove the tumor with minimal retraction of the spinal cord, additional laminectomy was performed. Various benefits derived from the use of IOUS were analyzed.

### Statistical analysis

All statistical analyses were performed using SPSS version 26.0 (IBM Corp., Armonk, New York, USA), and a P value < .05 was considered statistically significant. To compare the EQ-5D-5L and MMC between preoperative assessments and the last follow-up, a paired t-test was conducted.

### Ethics statement

Our institutional review board waived the requirement for informed consent and approved the study protocol and chart review (Approval No. H-2310-084-1475). All investigations were conducted in accordance with our institutional review board of guidelines and regulations.

## Results

The IOUS was used for 43 patients with spinal cord surgeries. By diagnosis, schwannoma was the most common with 15 patients (35%), followed by cavernous angioma (CA) with 10 patients (23%) and ependymoma (EPN) with 7 patients (16%). The mean age of the patients was 46.8 years, and there were 22 males (51.2%). Tumors were most commonly located in the cervical spine with 22 cases (51.2%). The mean of preoperative EQ-5D-5L was 0.624 and the mean of preoperative MMC was 1.7 (Table 1).

### Clinical and oncologic outcomes

The mean operation time was 237.2 minutes, and gross total resection (GTR) was achieved in 40 patients (93.0%). In CA patients, motor deterioration after surgical resection was observed in 3 patients (30% of 10 patients), and CSF leakage occurred in 1 patient (10% of 10 patients). In EPN patients, postoperative motor deterioration was confirmed in 1 patient (14.3% of 7 patients), and in subependymoma (SUBEP) patient, it was confirmed in 1 patient (100%). The mean length of hospital stay (LOS) for the patients was 6.1 days, and the mean follow-up duration was 5.6 months post-surgery. At the last follow-up imaging, suspicious recurrence was detected in one CA patient (10% of 10 patients) and one hemangioblastoma (HB) patient (33.3% of 3 patients). Consequently, an additional MR imaging follow-up was scheduled for one year later. The mean EQ-5D-5L score at the last follow-up was 0.693 (P = .140; compared

**Table 1. Baseline characteristics.**

| Diagnosis | N | Age[†] (year) | Sex (M/F) | Spinal level | Preoperative EQ-5D-5L[†] | Preoperative MMC[†] |
|---|---|---|---|---|---|---|
| SWN | 15 | 41.7 | 9/6 | Cervical: 9<br>Cervicothoracic: 1<br>Thoracic: 4<br>Lumbar: 1 | 0.654 | 1.4 |
| CA | 10 | 40.9 | 5/5 | Cervical: 4<br>Thoracic: 6 | 0.625 | 2.1 |
| EPN | 7 | 48.7 | 4/3 | Cervical: 1<br>Cervicothoracic: 1<br>Thoracic: 1 | 0.604 | 1.7 |
| HB | 3 | 40.0 | 2/1 | Cervical: 1<br>Thoracic: 2 | 0.758 | 1.7 |
| MNG | 3 | 45.3 | 0/3 | Cervical: 1<br>Thoracic: 2 | 0.434 | 2.0 |
| LGG | 1 | 64 | 0/1 | Thoracic: 1 | 0.751 | 1 |
| SUBEP | 1 | 47 | 0/1 | Cervical: 1 | 0.772 | 2 |
| SFT | 1 | 43 | 0/1 | Cervical: 1 | 0.466 | 2 |
| MPE | 1 | 36 | 1/0 | Lumbar: 1 | 0.463 | 2 |
| AC | 1 | 40 | 1/0 | Thoracic: 1 | 0.542 | 2 |
| Total | 43 | 46.8 ± 13.9 (21–80)[‡] | 22/21 | Cervical: 22<br>Cervicothoracic: 2<br>Thoracic: 17<br>Lumbar: 2 | 0.624 ± 0.227 (-0.310–0.904) [‡] | 1.7 ± 0.6 (1–4) [‡] |

N; number of patients, M; male, F; female, EQ-5D-5L; EuroQol 5-Dimension 5-Level, MMC; modified McCormick scale, SWN; schwannoma, CA; cavernous angioma, EPN; ependymoma, HB; hemangioblastoma, MNG; meningioma, LGG; low grade glioma, SUBEP; subependymoma, SFT; solitary fibrous tumor, MPE; myxopapillary ependymoma, AC; arachnoid cyst

[†]A variable is presented as a mean

[‡]A variable is presented as a mean ± standard deviation (range)

with preoperative EQ-5D-5L), and the mean MMC at the last follow-up was 1.5 (P = .002; compared with preoperative MMC) (Table 2).

## Utilization of intraoperative ultrasonography in spinal cord surgery

The advantages of IOUS identified in this study are presented in Table 2. IOUS was used to verify the adequate exposure of the extent and location of the lesion before the dural opening in 42 patients (97.7%). Additional laminectomy was performed to ensure adequate exposure of the tumor in one patient. The confirmation of lesion location was especially useful for a deep-seated intramedullary (IM) tumor, which was not evident even after the opening of dura in 13 patients. Before myelotomy, confirming the location of the IM tumor helped in determining the extent of the myelotomy. In one patient, it facilitated the identification of a tiny IM tumor that remained undetected in preoperative MR imaging (Case 3). After the tumor removal, the surgical field was checked with IOUS to detect any residual mass, and if present, additional tumor resection was performed to achieve GTR in three patients (7.0%). Especially in IM tumors, IOUS was used to check free pulsation of the spinal cord and absence of hematoma after resection of the tumor, and after closure of myelotomy and dura in 23 patients (53.5%). All of them, normal spinal cord pulsation was observed, and no hematoma was detected. For extradural (ED) neurogenic tumors, preoperative MR imaging may not always provide clear differentiation regarding the coexistence of intradural (ID) tumor with ED tumor. The

**Table 2. Clinical and oncologic outcomes classified by diagnosis.**

| Diagnosis | OP time (min) [†] | DOR | MEP attenuation | Complication | LOS (day) [†] | FU duration (min) [†] | Last FU tumor status | Last FU EQ-5D-5L [†] | Last FU MMC [†] |
|---|---|---|---|---|---|---|---|---|---|
| SWN | 205 | GTR: 13 non-GTR: 2 | 2 | | 4.3 | 4.9 | Stable: 15 | 0.762 | 1.3 |
| CA | 236 | GTR: 10 | 3 | Weakness: 3 CSF leakage: 1 | 6.1 | 4.6 | Stable: 9 Suspicious recur: 1 | 0.695 | 1.7 |
| EPN | 318 | GTR: 7 | 1 | Weakness: 1 | 11.4 | 7.4 | Stable: 7 (adj RT: 1) | 0.610 | 1.7 |
| HB | 215 | GTR: 3 | 1 | | 5 | 6.8 | Stable: 2 Suspicious recur: 1 | 0.666 | 1.3 |
| MNG | 218 | GTR: 3 | 0 | | 5 | 6.8 | Stable: 3 | 0.776 | 1 |
| LGG | 195 | non-GTR: 1 | NA | | 4 | 6 | Stable: 1 (adj RT: 1) | 0.738 | 1 |
| SUBEP | 325 | GTR: 1 | 0 | Weakness: 1 | 6 | 18.5 | Stable: 1 | 0.35 | 2 |
| SFT | 215 | GTR: 1 | 0 | | 5 | 1.1 | Stable: 1 (adj RT: 1) | 0.38 | 1 |
| MPE | 275 | GTR: 1 | 0 | | 4 | 0.9 | Stable: 1 | 0.862 | 1 |
| AC | 230 | GTR: 1 | 1 | | 6 | 1 | Stable: 1 | 0.506 | 2 |
| Total | 237.2 ± 73.7 (105–430) [‡] | GTR: 40 non-GTR: 3 | Attenuation: 8 non-Attenuation: 32 NA: 1 | Weakness: 5 CSF leakage: 1 | 6.1 ± 6.0 (4–43) [‡] | 5.6 ± 5.9 (0.7–19.6) [‡] | Stable: 41 Recur: 2 | 0.693 ± 0.188 (-0.303–0.904) [‡] | 1.5 ± 0.5 (1–4) [‡] |

OP; operation, DOR; degree of resection; MEP; motor-evoked potential, LOS; length of hospital stay, FU; follow-up, EQ-5D-5L; EuroQol 5-Dimension 5-Level, MMC; modified McCormick scale, SWN; schwannoma, GTR; gross total resection, CA; cavernous angioma, CSF; cerebrospinal fluid, EPN; ependymoma, adj RT; adjuvant radiotherapyHB; hemangioblastoma, MNG; meningioma, NA; not applicable, LGG; low grade glioma, SUBEP; subependymoma, SFT; solitary fibrous tumor, MEP; myxopapillary ependymoma, AC; arachnoid cyst

[†] A variable is presented as a mean

[‡] A variable is presented as a mean ± standard deviation (range)

presence or absence of a co-existing ID tumor was checked with IOUS after resection of the ED tumor, and the absence of an ID tumor allowed for the prevention of unnecessary additional dural openings in three patients (7.0%). In tumor encasing the vertebral artery (VA), it was possible to assess the preservation of VA flow before, during, and after tumor removal using color Doppler mode in one patient (2.3%). Finally, in a patient (2.3%) with an arachnoid cyst and ventral spinal cord herniation, the ventral dural defect could be identified using IOUS and subsequently repaired.

## Illustrative case 1

A 39-year-old female patient, who had been experiencing right lower limb weakness for the past 10 years, underwent MR imaging which revealed T9-10 IM lesions, suggesting CAs (Fig 1A). A laminectomy was intended at T9 and T10, but the lesions were not fully visualized with IOUS. A fluoroscopy was brought in again and the image revealed that laminectomy was done at T8 and T9. Consequently, the lower margin of the CAs was not identified and an additional laminectomy at T10 was performed before opening the dura. Subsequent verification with IOUS confirmed the complete exposure of the CAs' extent. (Fig 1B). On the IOUS, the lesions were located at the right ventral side of the spinal cord. A posterolateral myelotomy was performed to expose the lesion. Initially, the lesion on the cranial side was removed (Fig 1C).

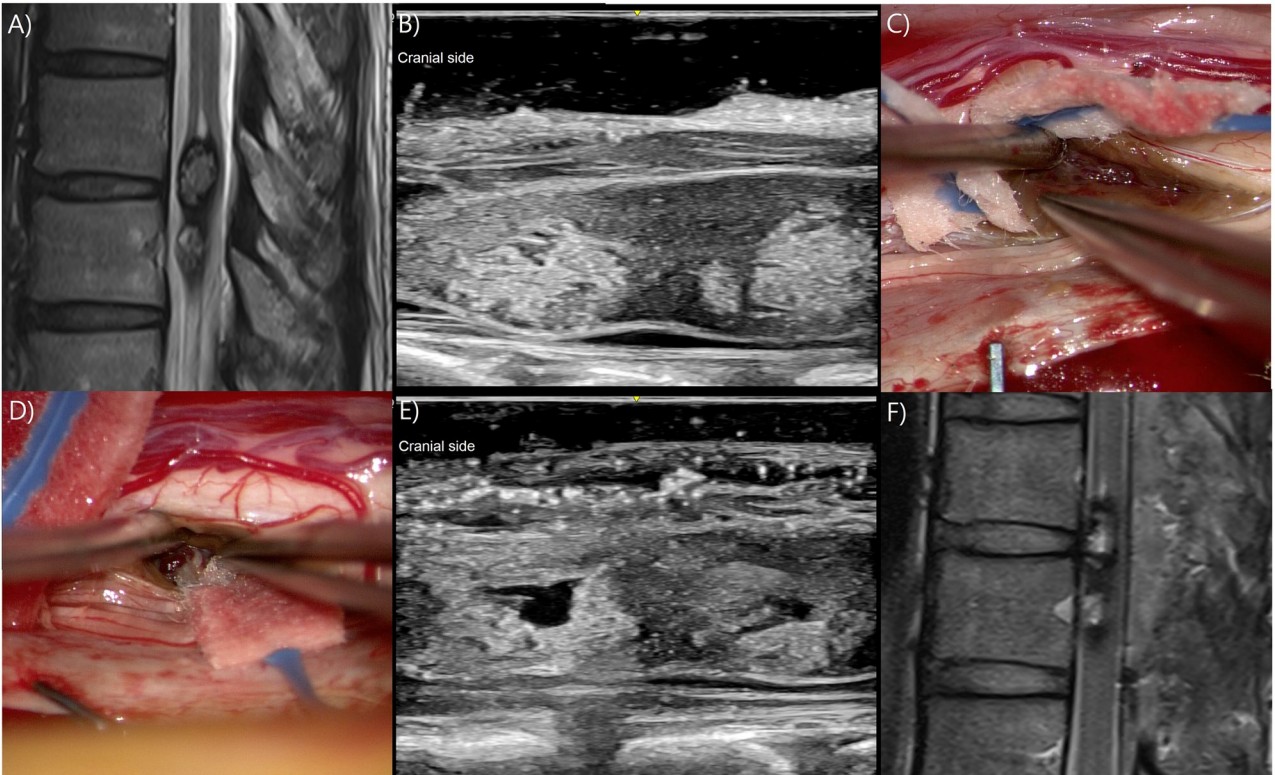

**Fig 1. Illustrative case 1.** A) T9-10 intramedullary (IM) lesions are identified on magnetic resonance (MR) imaging. B) Before the dural opening, intraoperative ultrasonography (IOUS) revealed hyperechoic IM lesions situated at the right ventral side of the spinal cord. C) Following the posterolateral myelotomy, the lesion on the cranial side was first identified, and the cavernous angioma (CA) exhibiting a mulberry appearance was excised. D) Subsequently, the lesion located on the caudal side was identified and the CA was removed. E) Following the complete removal of the CAs, IOUS was used to confirm the absence of any residual lesions and intracavitary hematoma. F) On postoperative MR imaging, it was confirmed that the CAs had been grossly totally resected.

IOUS accurately localized the caudal lesion, facilitating the identification of the lesion's axial location and assisting in determining the extent of the myelotomy. After the complete removal of the second CA (Fig 1D), GTR and no hematoma in the surgical cavity were confirmed using IOUS (Fig 1E). The final pathological diagnosis was CA. Postoperative MR imaging showed complete removal of CAs (Fig 1F). Postoperatively, the patient experienced a temporary worsening of right ankle dorsiflexion weakness, which gradually improved to an ambulatory state 3 days after surgery, allowing for discharge.

## Illustrative case 2

A 60-year-old female patient presented with posterior neck pain that started two months ago. MR imaging revealed a lesion at C3-4 with extraforaminal extension to the left side (Fig 2A). A schwannoma was suspected. A left C3-4 hemilaminectomy and partial facet resection were performed to expose the tumor. Subsequently, the epineurium and perineurium were incised along the direction of the tumor, and the tumor was dissected utilizing a sub-perineurial dissection (Fig 2B). The tumor was removed by pulling out the tumor from the dural sac and the ID tumor was removed. Since the IOUS confirmed the absence of any residual ID portion (Fig 2C), no further dural opening was performed. The left VA, which was encased and displaced by the tumor, exhibited intact blood flow in color Doppler mode after tumor removal

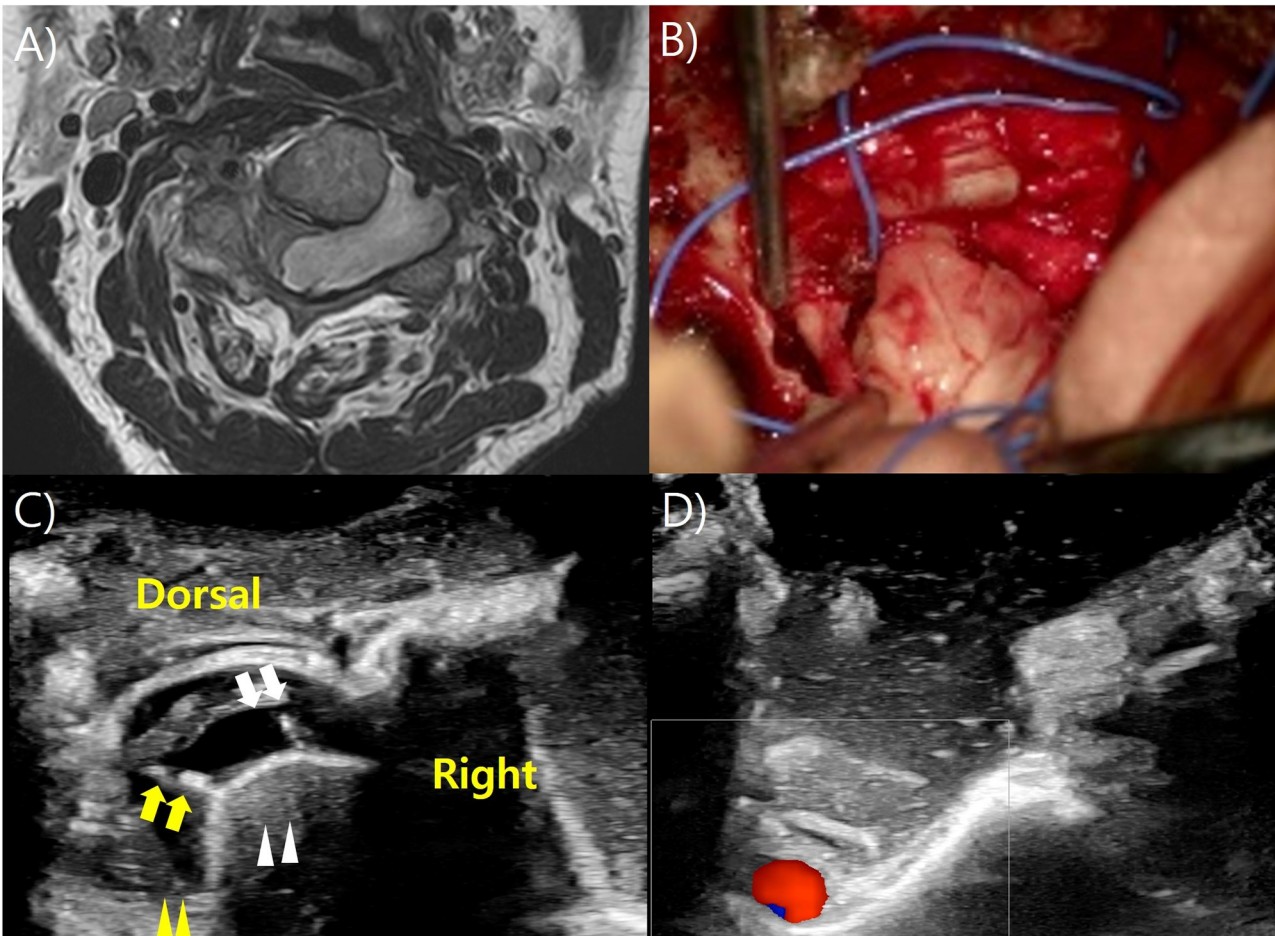

**Fig 2. Illustrative case 2.** A) Magnetic resonance (MR) imaging revealed a C3-4 intradural-extramedullary (IDEM) lesion with extraforaminal extension, which is encasing and displacing the left vertebral artery (VA). B) After C2 hemi laminectomy, the lesion was removed from the extradural side through sub-perineurial dissection, and the intradural (ID) lesion was extracted using the pull-out method through the dural sleeve. C) Intraoperative ultrasonography (IOUS) was performed through a window of laminectomy and it revealed the dentate ligament (yellow arrow), dorsal root (white arrow), ventral root (yellow arrowhead), and spinal cord (white arrowhead), confirming the absence of any residual ID tumor. D) Following the removal of the tumor, the integrity of blood flow in the VA was confirmed using color Doppler mode.

(Fig 2D). Postoperative MR imaging confirmed the GTR of the tumor, and the final pathological diagnosis was schwannoma. The patient was discharged without any new neurological deficits following the surgery.

### Illustrative case 3

A 30-year-old male patient, previously diagnosed with Von Hippel-Lindau syndrome visited with weakness in the right hand. MR imaging revealed a single IM enhancing nodule and a syrinx at the C7 level (Fig 3A). Clinically, a hemangioblastoma was suspected. In the angiogram of the right deep cervical artery, a single hypervascular nodule was identified at the C7 level (Fig 3B). In the angiogram of the left subclavian artery, no additional hypervascular nodules were identified (Fig 3C). Upon performing a laminectomy at C6 and C7, and before the dural opening, IOUS confirmed the presence of a nodule located on the right side of the spinal cord, as identified in MR imaging (Fig 3D). However, an additional tiny nodule, not detected

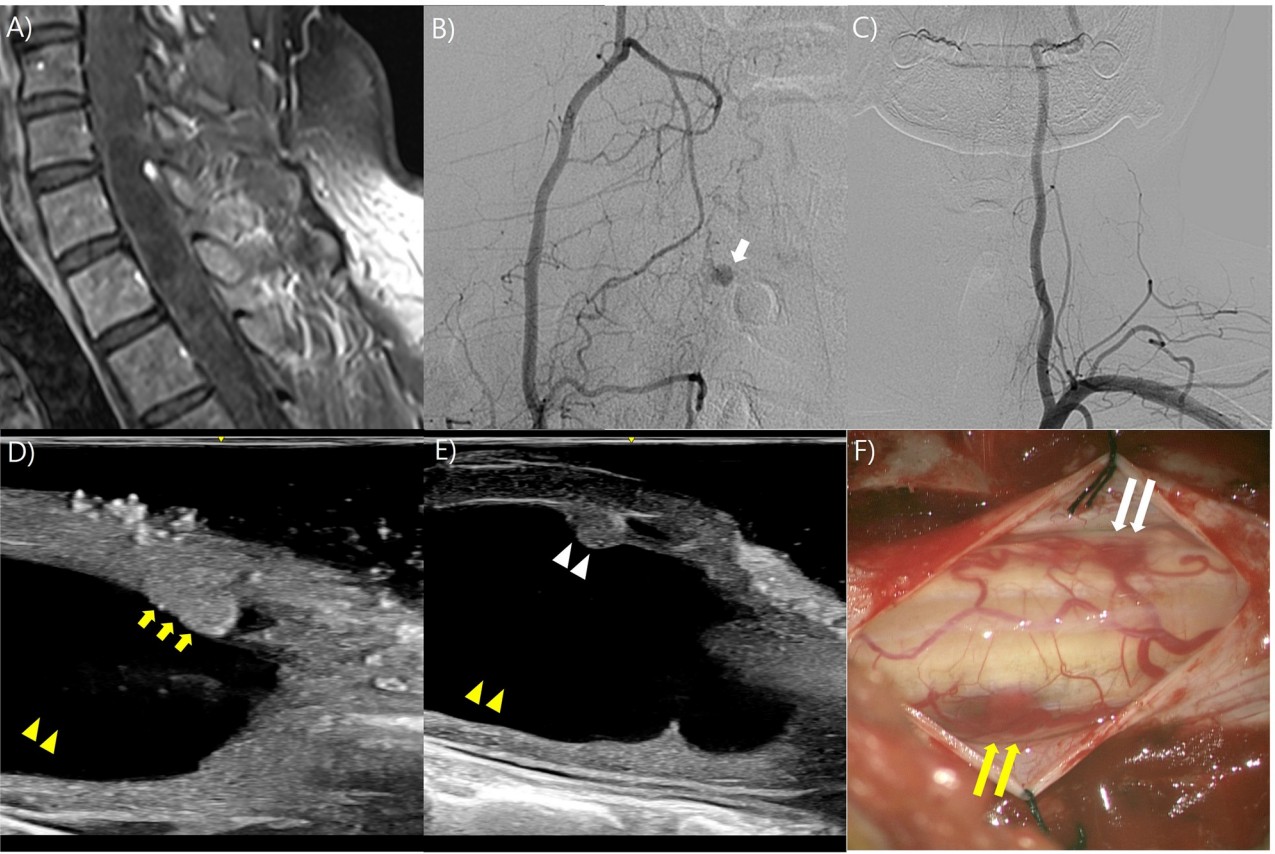

**Fig 3. Illustrative case 3.** A) Magnetic resonance (MR) imaging revealed an intramedullary (IM) enhancing nodule and a syrinx at the C7. B) In the right deep cervical artery angiogram, a hypervascular nodule (white short arrow) was identified at the C7. C) In the left subclavian artery angiogram, no additional hypervascular nodules were identified. D) Before the dural opening, intraoperative ultrasound (IOUS) was employed to confirm the presence of a syrinx (yellow arrowhead), and the nodule (yellow short arrow) on the right side of the spinal cord, previously identified in magnetic resonance (MR) imaging, was observed. E) An additional tiny nodule (white arrowhead), not detected in MR imaging and angiogram, was identified on the left side of the spinal cord at the boundary of the syrinx (yellow arrowhead) and the spinal cord. F) After the dural opening, both nodules (yellow long arrow; previously detected nodule in MR imaging and angiogram, white long arrow; newly detected nodule in IOUS) identified by IOUS were visualized.

in MR imaging and angiogram, was identified on the left side of the spinal cord (Fig 3E). Following the dural opening, both nodules identified by IOUS were visualized and completely resected. The final pathological diagnoses of two nodules were hemangioblastomas (Fig 3F). The patient exhibited improvement in the weakness of the right hand and was discharged without any new neurological deficits.

### Illustrative case 4

A 39-year-old male patient presented with gait disturbance and sensory loss in the right leg that had developed over the past six months. MR imaging revealed a possible T4-5 arachnoid cyst with spinal cord herniation (Fig 4A). After laminectomy at T4 and T5, IOUS revealed that a dorsal cystic lesion was compressing the spinal cord. Upon dural opening, the arachnoid cyst was bulged out (Fig 4B). After the removal of the arachnoid cyst, an inspection of the spinal cord using IOUS revealed a dural defect on the ventral side of the herniated spinal cord (Fig 4C). The herniated spinal cord was dissected out from the dural defect (Fig 4D) and the

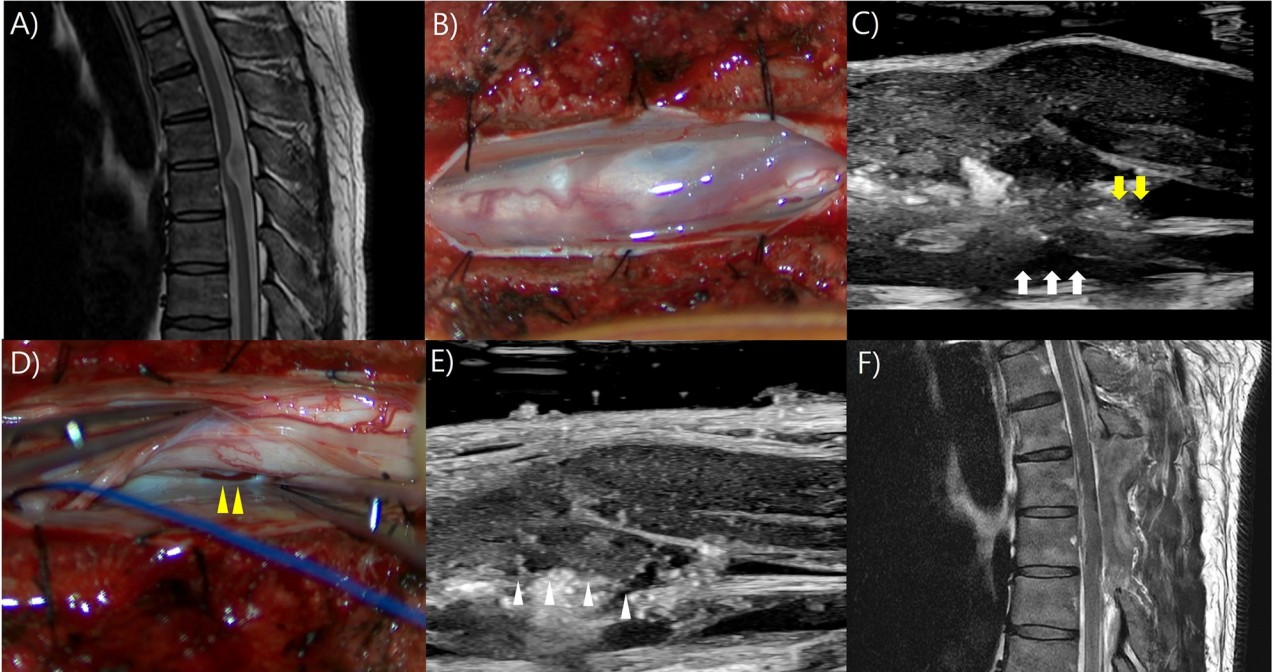

**Fig 4. Illustrative case 4.** A) On magnetic resonance (MR) imaging, a possible arachnoid cyst with spinal cord herniation at T4-5 is identified. B) Upon performing the dural opening, an arachnoid cyst was identified and subsequently removed. C) Intraoperative ultrasonography (IOUS) revealed a ventral dural defect (yellow arrow) and a herniated spinal cord (white arrow). D) Upon inspection at the location identified by IOUS, a ventral dural defect (yellow arrowhead) was observed under microscopic view. E) After repairing the dural defect with artificial dura, the reduction of the herniated spinal cord (white arrowhead) was confirmed using IOUS. F) Postoperative MR imaging showed resolution of the spinal cord herniation.

dural defect was repaired using artificial dura. After the closure of the dura, IOUS showed a restored herniated spinal cord (Fig 4E) and postoperative MR imaging showed resolution of the spinal cord herniation (Fig 4F). The pathological diagnosis was an arachnoid cyst. Postoperatively, the patient experienced a temporary exacerbation of weakness in left ankle dorsiflexion, which gradually improved to an ambulatory state 3 days after surgery, leading to discharge.

## Discussion

In spinal cord surgeries, the use of IOUS is not widely adopted due to the low quality of imaging, spatial constraints of the surgical field, and underrepresentation of its usability [5, 21]. There have been many reports of utilizing IOUS in the diagnosis and treatment of various spinal diseases, such as spinal trauma, degenerative spinal diseases, congenital spinal disorders, and spinal cord tumors [22–25]. In this study, we utilized IOUS in spinal cord surgeries. After laminectomy, ultrasonography through the dura and the spinal cord allowed real-time assessment of the anatomical relationship between the tumor and surrounding structures. This facilitated the patient-specific surgical strategies aimed at maximizing tumor resection while minimizing neurological deficits through a customized approach.

At our institution, the use of IOUS during spinal cord tumor surgeries was initiated in November 2021. In these surgeries, when the dura was opened and additional bone work was required, the bone work was done with a high risk of nerve damage due to the exposed spinal cord. This is particularly notable in cases of schwannomas, where the corresponding authors

have observed slight positional shifts from the prone position that are not evident in preoperative MR images. These shifts sometimes necessitate additional laminectomy to adequately expose the surgical field. Moreover, in the treatment of ED-ID tumors, the ED portion is first removed, followed by the ID portion using the pull-out method. Subsequently, to ensure no residual tumor remains, an additional dural opening is often required. This extra step not only extends the surgery time but also elevates the risk of cerebrospinal fluid (CSF) leakage. Additionally, there have been instances where, despite the successful removal of a VA encasing schwannoma without damaging the VA, postoperative MR imaging revealed compromised flow in the ipsilateral VA. To mitigate these issues and risks encountered during surgery, our institution began utilizing IOUS in spinal cord tumor surgeries.

The usage of IOUS in the present study is summarized in Table 3. The list of usage was limited to the present cases, but the usage could be variably extended according to the needs of the surgeon. The primary advantage of IOUS was the real-time exact localization of the lesion. By confirming the extent and location of the lesion using IOUS before dural opening and myelotomy, the size of the dural opening and myelotomy required for tumor exposure could be optimized [26, 27]. If the exposure of the lesion is deemed insufficient, additional bone work can be performed before the dural opening, which aids in minimizing the manipulation of neural structures [28–30]. As Case 1 showed, IOUS allowed an extension of laminectomy due to insufficient exposure or wrong level laminectomy before opening the dura. Particularly, when removing deep-seated small lesions, IOUS enabled the precise real-time determination of the location and depth, facilitated accurate myelotomy, and minimized the extent of myelotomy, thereby reducing additional injury to neural tissue (Case 1). When ID tumors extend extraforaminal location, the presence of residual ID tumor after removal of the tumor should be checked out, because the tumor was usually removed by pulling out the tumor through the dural sleeve (Case 2). Exploratory opening of the dura may be the surest way, but it needs additional surgical time and procedures. The residual ID tumor could be non-invasively checked out using IOUS, thereby unnecessary additional dural openings could be avoided. Utilizing Doppler mode enables the assessment of vascular structures within or around the tumor [31]. The VA, encased and displaced by the tumor, was identified using IOUS, and it allowed the removal of the tumor without damaging the VA (Case 2). After tumor removal, the intact blood flow of the VA was confirmed using color Doppler mode. Additionally, IOUS facilitates the real-time identification of residual or hidden tumors (Case 3), aiding in total resection of tumors [32]. IOUS also assists in confirming the absence of hematoma within the surgical cavity [33]. In this study, a particularly noteworthy benefit of IOUS was its utility in a patient with spinal cord herniation through a ventral dural defect (Case 4). The real-time identification of

**Table 3. Advantage of intraoperative ultrasonography in this study.**

| Role of intraoperative ultrasonography | N (%) |
|---|---|
| Verification of tumor extent and location, especially deep-seated intramedullary tumor | 42[†] (97.7) |
| Detection and additional removal of residual or hidden tumor | 3 (7.0) |
| Identification of the absence of hematoma in the surgical cavity | 23 (53.5) |
| Confirming the absence of intradural lesion to prevent additional dural incision | 3 (7.0) |
| Verification of vertebral artery flow intactness | 1 (2.3) |
| Detection of the ventral dural defect | 1 (2.3) |

N; number of patients

[†]One patient; additional laminectomy for adequate exposure of tumor, thirteen patients; deep-seated intramedullary tumors

the dural defect's location with IOUS enabled the defect repair with minimal spinal cord manipulation, thereby preventing cord injury.

IOUS could be used for the intraoperative diagnosis. There are several reports characterizing the IOUS features of spinal cord tumors [31, 34, 35]. Although we utilized IOUS mostly for localization of the pathology, the reported IOUS appearances of spinal cord tumors may be utilized for a specific diagnosis (Table 4). Representative IOUS imaging based on diagnosis is shown in Fig 5. Sonographic features were retrospectively reviewed by radiologist. While specific sonographic features vary among different spinal cord tumors and modes of ultrasonography machines, spinal cord tumors generally exhibit hyperechoic characteristics [31, 34, 35]. Excluding astrocytoma, which exhibits infiltrative growth, most spinal cord tumors are discernible from the normal spinal cord by a distinct border [31, 34, 35]. Additionally, when contrast is employed, the boundary between spinal cord tumors and the normal spinal cord can be more clearly delineated [31, 34–36]. Han et al. prospectively recruited 14 patients with IM tumors to evaluate the potential role of contrast-enhanced ultrasonography [36]. They reported that contrast-enhanced ultrasonography not only delineates the tumor boundaries more clearly but also provides potential information about the tumor's perfusion status [36]. Vetrano et al. reported that in patients with hemangioblastoma, the exact location of the nodule was determined using standard B-mode ultrasonography, Doppler mode, and contrast-enhanced ultrasonography. This allowed for the complete removal of the tumor with minimal spinal cord manipulation [37]. The literature reporting on oncologic outcomes associated with the use of IOUS is even scarcer. Wael et al. [38] reported that when the IOUS showed distinct boundaries and a smooth shape of a spinal cord tumor, it suggested a higher rate of complete tumor resection and a better functional outcome.

## Limitation

This study has several limitations. Firstly, this study was conducted at a single institution with a relatively small number of subjects. Although we presented the usability of IOUS, the listed usability was not all and it could be variably extended by surgeons. Secondly, due to the small number of patients and relatively short follow-up period, the association between clinical/oncologic outcomes in patients who used IOUS versus those who did not use IOUS was not analyzed. A prospective study may be required to verify the beneficial effect of IOUS on clinical/oncologic outcomes. Thirdly, the study did not evaluate the delay in surgical time and the increase in medical costs associated with the use of IOUS. Fourthly, basic analysis of IOUS features can be conducted by surgeons without specialized training; however, the capacity for professional interpretation, particularly for use in deferential diagnosis of tumors, collaboration with a radiologist may be necessary. Finally, IOUS was available only after laminectomy. If the technology of IOUS improved, a trans-laminar IOUS may enhance the benefit of IOUS in conducting spinal cord surgery. Nonetheless of those limitations, this study showed the usability of IOUS with representative cases and this information may be helpful for general use of IOUS.

## Conclusion

In spinal cord surgery, IOUS not only aids in providing real-time imagings to confirm the tumor's extent and location, but also assists in identifying residual or hidden lesions, vascular injuries, and dural defects. Therefore, the utilization of IOUS in spinal cord surgery could be a valuable tool, not only in reducing the risk of damage to normal neural and vascular structures but also in optimizing tumor resection.

**Table 4. Sonographic feature of spinal cord disease in other literature.**

| Study | EPN | AS | CA | HB | MNG | SWN | SUBEP | SFT | MPE | AC |
|---|---|---|---|---|---|---|---|---|---|---|
| Prada et al. [27] (2014) | hyperechoic, circumscribed, homogenous, small/microcystic polar area | hyperechoic, fine, granular, blurred margin | highly hyperechoic, microcystic, microcalcification, less circumscribed, more nodular | hyperechoic, nodular, homogenous, perilesional cyst, macrocytic | mild hyperechoic, well-circumscribed, homogenous | mild hyperechoic, circumscribed, less homogenous than MNG | | | | |
| Toktas et al. [24] (2013) | hyperechoic, mixed | isoechoic, hypoechoic | mixed, hypoechoic | hyperechoic nodule, less echoic cyst | hyperechoic | hyperechoic, isoechoic, hypoechoic | | | | |
| Barkley et al. [28] (2021) | isoechoic, enhanced | isoechoic, non-enhanced | | hyperechoic, isoechoic, enhanced | | | isoechoic, non-enhanced | | | |
| Our study | Hyperechoic; circumscribed, homogenous, microcysts | isoechoic, cord swelling, blurred margin | hyperechoic, circumscribed, microcyst | isoechoic, nodular, homogenous, cyst | hyperechoic, circumscribed, heterogenous | hypoechoic, circumscribed, heterogenous, cyst | hypoechoic, cyst | hypoechoic, cyst | hypoechoic, homogenous, cyst | anechoic |

EPN; ependymoma, AS; astrocytoma, CA; cavernous angioma, HB; hemangioblastoma, MNG; meningioma, SWN; schwannoma, SUBEP; subependymoma, SFT; solitary fibrous tumor, MPE; myxopapillary ependymoma, AC; arachnoid cyst

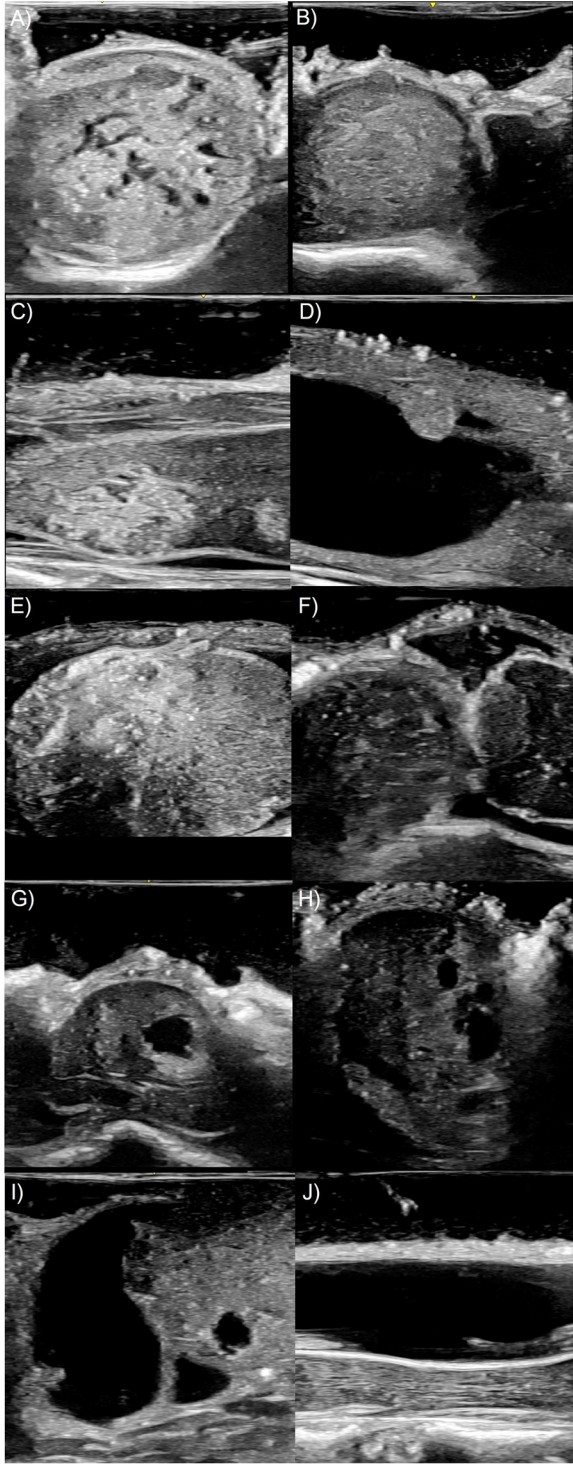

**Fig 5. Representative findings of intraoperative ultrasonography.** A) Ependymoma. In intraoperative ultrasonography (IOUS), the lesion appears hyperechoic, circumscribed, and homogeneous, with accompanying microcysts observed. B) Astrocytoma. In IOUS, the lesion is isoechoic, exhibiting cord swelling and a blurred margin. C) Cavernous angioma. In IOUS, the lesion appears hyperechoic and circumscribed, with accompanying microcysts observed. D) Hemangioblastoma. In IOUS, an isoechoic and homogeneous nodule is observed along with a cyst. E) Meningioma. In IOUS, the lesion appears hyperechoic, circumscribed, and heterogenous. F) Schwannoma. In IOUS, the lesion is hypoechoic, circumscribed, heterogeneous, and is observed with an accompanying cyst. G) Subependymoma. In IOUS, the lesion is hypoechoic and is accompanied by a cyst. H) Solitary fibrous tumor. In IOUS,

the lesion is hypoechoic and is accompanied by cysts I) Mixopapillary ependymoma. In IOUS, the lesion appears hypoechoic, homogeneous, and is associated with accompanying cysts. J) Arachnoid cyst. In IOUS, the lesion is observed as anechoic.

## Supporting information

**S1 File.**
(XLSX)

## Author Contributions

**Conceptualization:** Chi Heon Kim.

**Data curation:** Hangeul Park, Jun-Hoe Kim, Chang-Hyun Lee.

**Formal analysis:** Ji-hoon Kim.

**Funding acquisition:** Chi Heon Kim.

**Investigation:** Hangeul Park, Sum Kim, Young-Rak Kim.

**Resources:** Woo-Young Jo, Hyongmin Oh, Hee-Pyoung Park.

**Supervision:** Kyung-Tae Kim, John M. Rhee.

**Writing – original draft:** Hangeul Park.

**Writing – review & editing:** John M. Rhee, Hyongmin Oh, Hee-Pyoung Park, Chi Heon Kim.

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
