## [Decision Letter · Decision Letter 0]

20 May 2024

PONE-D-24-13339The utility of intraoperative ultrasonography for spinal cord surgeryPLOS ONE

Dear Dr. Kim,

Thank you for submitting your manuscript to PLOS ONE. After careful consideration, we feel that it has merit but does not fully meet PLOS ONE’s publication criteria as it currently stands. Therefore, we invite you to submit a revised version of the manuscript that addresses the points raised during the review process.

We look forward to receiving your revised manuscript.

Kind regards,

Barry Kweh

Academic Editor

PLOS ONE

“This study was supported by Ministry of National Defence of Republic of Korea (800-20230466) and Doosan Yonkang foundation (800-20210527). This study was also supported by grant (30-2023-0120) from the Seoul National University Hospital research fund.”

3. In the online submission form you indicate that your data is not available for proprietary reasons and have provided a contact point for accessing this data. Please note that your current contact point is a co-author on this manuscript. According to our Data Policy, the contact point must not be an author on the manuscript and must be an institutional contact, ideally not an individual. Please revise your data statement to a non-author institutional point of contact, such as a data access or ethics committee, and send this to us via return email. Please also include contact information for the third party organization, and please include the full citation of where the data can be found.

Additional Editor Comments:

A well written manuscript that requires a broader discussion of the literature and other adjuncts in spinal surgery. A tabulated and written summary of similar studies examining the utility and efficacy of US in spinal cord surgery, as well as other adjuncts such as neuromonitoring would greatly benefit the audience.

Reviewers' comments:

Reviewer's Responses to Questions

**Comments to the Author**

1. Is the manuscript technically sound, and do the data support the conclusions?

Reviewer #1: Yes

Reviewer #2: Yes

2. Has the statistical analysis been performed appropriately and rigorously? 

Reviewer #1: Yes

Reviewer #2: N/A

3. Have the authors made all data underlying the findings in their manuscript fully available?

Reviewer #1: Yes

Reviewer #2: Yes

4. Is the manuscript presented in an intelligible fashion and written in standard English?

Reviewer #1: Yes

Reviewer #2: Yes

5. Review Comments to the Author

Reviewer #1: In this study the authors advocate for the use of intraoperative US in spinal cord surgery, specifically in tumor cases. They present their series of 43 patients in whom ultrasound was utilized to confirm the limits of the tumor before myelotomy and give 4 detailed examples of illustrative cases. The manuscript is easy to read without getting the reader tired. There are not that many case series in the literature so it might have some potential. However, overall the study needs several major revisions to qualify for publication in PLOS ONE.

Specific comments:

• Was an IRB required for the study? Please elaborate.

• The results are insightful, but too few variables have been included. I encourage the authors to include more variables, like intraoperative time, length of stay, postop destination, etc.

• Is degree of resection base on the MRI? Please clarify in methods

• In case 3, why do the authors think that the US had more utility in detecting “a tiny IM tumor that remained undetected in preoperative MR”?

• Do the authors have cases where IOUS was not used? It would be very helpful to the reader to have a control group to compare degree of resection, outcome scales, etc. Otherwise the utility of the US intraoperatively is limited to the exact localization of the tumor before opening the dura.

• Please provide a paragraph with relevant history on how IOUS started being during these cases in the past.

• The study doesn’t include oncology follow-up of the patients neither does it mention when the last follow up was. This paper is more like a technical report, and this should be indicated in the title/abstract.

• The degree of resection should be assessed by two independent reviewers, potentially one from radiology.

Thank you

Reviewer #2: The authors provide a nice case series of patients who underwent spinal surgery for a variety of lesions with intraoperative ultrasound utilized. intraoperative ultrasound helped the authors to localize the lesions, confirm extent of resection and confirm lack of local or regional complications. although this is not a new technique and is fairly widely used, the authors seem to be particularly proficient with its use. the authors remind the readership how straightforward, natural and helpful intraoperative ultrasound can be. perhaps many of us need to using it more frequently. if the authors hope to encourage our use of intraoperative ultrasound, they have done so with this case series.

6. PLOS authors have the option to publish the peer review history of their article (what does this mean?). If published, this will include your full peer review and any attached files.

Reviewer #1: **Yes: **Stavros Matsoukas

Reviewer #2: No

---

## [Author Response · Author response to Decision Letter 0]

26 May 2024

Response to reviewers

We sincerely appreciate the constructive feedback and discussions from the reviewers, and we have revised our paper to the best of our ability to address their comments. Our point-by-point responses are listed below.

Reviewer #1:

1. Was an IRB required for the study? Please elaborate

Thank you for your valuable comment. This retrospective study qualifies as minimal risk research, which poses the least possible risk to patients. As this study involved the retrospective collection and analysis of patient medical records, it utilized clinical data from patients for the research. In such cases, the Institutional Review Board (IRB) at Seoul National University Hospital recommends that all research protocols undergo an ethical review based on the 7th revision of the Declaration of Helsinki and the domestic Bioethics and Safety Act.1,2 Therefore, in accordance with the IRB recommendations, this study also obtained approval from the IRB. 

1 World Medical Association. World Medical Association Declaration of Helsinki: Ethical Principles for Medical Research Involving Human Subjects. JAMA. 2013;310(20):2191–2194. doi:10.1001/jama.2013.281053

2 https://elaw.klri.re.kr/kor_service/lawView.do?hseq=52559&lang=ENG

2. The results are insightful, but too few variables have been included. I encourage the authors to include more variables, like intraoperative time, length of stay, postop destination, etc.

Thank you for your considerate comment. To make the findings of our study more insightful, as you suggested, we added several variables to the analysis. We have added the following variables to the analysis to enhance the insightfulness of our study: preoperative patient-reported functional status (EQ-5D-5L), operation time, motor-evoked potential (MEP) attenuation, length of stay, follow-up duration, last follow-up tumor status, and last follow-up EQ-5D-5L. However, the duration of intraoperative ultrasonography (IOUS) use during surgery was not measured separately; thus, it was not included, although it did not exceed five minutes. Based on these additional variables, we have revised the Materials and Methods section titled "Patients” (page 6, line 115), “Statistical analysis” (page 8, line 147), Results section (page 8, line 160), Results section titled “Clinical and oncologic outcomes” (page 11, line 171), and Table 1 (page 10, line 164) as follows, and have added a new Table 2 (page 12, line 183) about clinical and oncologic outcomes.

Materials and Methods

Patients

Since November 2021, IOUS has been used in spinal cord surgery when the ultrasonography machine is available. Electric medical records were collected on age, sex, diagnosis, tumor location, preoperative patient-reported functional status (EuroQol 5-Dimension 5-Level; EQ-5D-5L)(12), preoperative McCormick scale(13), operation time, degree of resection (DOR), attenuation of motor-evoked potential, postoperative complications, length of hospital stay, follow-up duration, last follow-up tumor status, last follow-up EQ-5D-5L, and last follow-up MMC. The attenuation of MEP was assessed based on an all-or-none criterion.(14)

Statistical analysis

All statistical analyses were performed using SPSS version 26.0 (IBM Corp., Armonk, New York, USA), and a P value < .05 was considered statistically significant. To compare the EQ-5D-5L and MMC between preoperative assessments and the last follow-up, a paired t-test was conducted.

Results 

The IOUS was used for 43 patients with spinal cord surgeries. By diagnosis, schwannoma was the most common with 15 patients (35%), followed by cavernous angioma (CA) with 10 patients (23%) and ependymoma (EPN) with 7 patients (16%). The mean age of the patients was 46.8 years, and there were 22 males (51.2%). Tumors were most commonly located in the cervical spine with 22 cases (51.2%). The mean of preoperative EQ-5D-5L was 0.624 and the mean of preoperative MMC was 1.7 (Table 1)

Table 1 Baseline characteristics

Diagnosis N Age† (year) Sex (M/F) Spinal level Preoperative EQ-5D-5L† Preoperative MMC†

SWN 15 41.7 9/6 Cervical: 9

Cervicothoracic: 1

Thoracic: 4

Lumbar: 1 0.654 1.4

CA 10 40.9 5/5 Cervical: 4

Thoracic: 6 0.625 2.1

EPN 7 48.7 4/3 Cervical: 1

Cervicothoracic: 1

Thoracic: 1 0.604 1.7

HB 3 40.0 2/1 Cervical: 1

Thoracic: 2 0.758 1.7

MNG 3 45.3 0/3 Cervical: 1

Thoracic: 2 0.434 2.0

LGG 1 64 0/1 Thoracic: 1 0.751 1

SUBEP 1 47 0/1 Cervical: 1 0.772 2

SFT 1 43 0/1 Cervical: 1 0.466 2

MPE 1 36 1/0 Lumbar: 1 0.463 2

AC 1 40 1/0 Thoracic: 1 0.542 2

Total 43 46.8 ± 13.9 

(21 – 80)‡ 22/21 Cervical: 22

Cervicothoracic: 2

Thoracic: 17

Lumbar: 2 0.624 ± 0.227 

(-0.310 – 0.904) ‡ 1.7 ± 0.6 (1 – 4) ‡

N; number of patients, M; male, F; female, EQ-5D-5L; EuroQol 5-Dimension 5-Level, MMC; modified McCormick scale, SWN; schwannoma, CA; cavernous angioma, EPN; ependymoma, HB; hemangioblastoma, MNG; meningioma, LGG; low grade glioma, SUBEP; subependymoma, SFT; solitary fibrous tumor, MPE; myxopapillary ependymoma, AC; arachnoid cyst

†A variable is presented as a mean

‡A variable is presented as a mean ± standard deviation (range)

Clinical and oncologic outcomes

The mean operation time was 237.2 minutes, and gross total resection (GTR) was achieved in 40 patients (93.0%). In CA patients, motor deterioration after surgical resection was observed in 3 patients (30% of 10 patients), and CSF leakage occurred in 1 patient (10% of 10 patients). In EPN patients, postoperative motor deterioration was confirmed in 1 patient (14.3% of 7 patients), and in subependymoma (SUBEP) patient, it was confirmed in 1 patient (100%). The mean length of hospital stay (LOS) for the patients was 6.1 days, and the mean follow-up duration was 5.6 months post-surgery. At the last follow-up imaging, suspicious recurrence was detected in one CA patient (10% of 10 patients) and one hemangioblastoma (HB) patient (33.3% of 3 patients). Consequently, an additional MR imaging follow-up was scheduled for one year later. The mean EQ-5D-5L score at the last follow-up was 0.693 (P = .140; compared with preoperative EQ-5D-5L), and the mean MMC at the last follow-up was 1.5 (P =.002; compared with preoperative MMC) (Table 2).

Table 2. Clinical and oncologic outcomes classified by diagnosis

Diagnosis OP time (min) † DOR MEP attenuation Complication LOS (day) † FU duration (min) † Last FU tumor status Last FU EQ-5D-5L† Last FU MMC†

SWN 205 GTR: 13

non-GTR: 2 2 4.3 4.9 Stable: 15 0.762 1.3

CA 236 GTR: 10 3 Weakness: 3

CSF leakage: 1 6.1 4.6 Stable: 9

Suspicious recur: 1 0.695 1.7

EPN 318 GTR: 7 1 Weakness: 1 11.4 7.4 Stable: 7 (adj RT: 1) 0.610 1.7

HB 215 GTR: 3 1 5 6.8 Stable: 2

Suspicious recur: 1 0.666 1.3

MNG 218 GTR: 3 0 5 6.8 Stable: 3 0.776 1

LGG 195 non-GTR: 1 NA 4 6 Stable: 1 (adj RT: 1) 0.738 1

SUBEP 325 GTR: 1 0 Weakness: 1 6 18.5 Stable: 1 0.35 2

SFT 215 GTR: 1 0 5 1.1 Stable: 1 (adj RT: 1) 0.38 1

MPE 275 GTR: 1 0 4 0.9 Stable: 1 0.862 1

AC 230 GTR: 1 1 6 1 Stable: 1 0.506 2

Total 237.2 ± 73.7

(105 – 430) ‡ GTR: 40

non-GTR: 3 Attenuation: 8

non-Attenuation: 32

NA: 1 Weakness: 5

CSF leakage: 1 6.1 ± 6.0

(4 – 43) ‡ 5.6 ± 5.9

(0.7 – 19.6) ‡ Stable: 41

Recur: 2 0.693 ± 0.188

(-0.303 – 0.904) ‡ 1.5 ± 0.5

(1 – 4) ‡

OP; operation, DOR; degree of resection; MEP; motor-evoked potential, LOS; length of hospital stay, FU; follow-up, EQ-5D-5L; EuroQol 5-Dimension 5-Level, MMC; modified McCormick scale, SWN; schwannoma, GTR; gross total resection, CA; cavernous angioma, CSF; cerebrospinal fluid, EPN; ependymoma, adj RT; adjuvant radiotherapyHB; hemangioblastoma, MNG; meningioma, NA; not applicable, LGG; low grade glioma, SUBEP; subependymoma, SFT; solitary fibrous tumor, MEP; myxopapillary ependymoma, AC; arachnoid cyst

†A variable is presented as a mean

‡A variable is presented as a mean ± standard deviation (range)

Reference

12. Kim SH, Ahn J, Ock M, Shin S, Park J, Luo N, et al. The EQ-5D-5L valuation study in Korea. Qual Life Res. 2016;25(7):1845-52.

13. McCormick PC, Torres R, Post KD, Stein BM. Intramedullary ependymoma of the spinal cord. Journal of neurosurgery. 1990;72(4):523-32.

14. Jin SH, Chung CK, Kim CH, Choi YD, Kwak G, Kim BE. Multimodal intraoperative monitoring during intramedullary spinal cord tumor surgery. Acta Neurochir (Wien). 2015;157(12):2149-55.

3. Is degree of resection base on the MRI? Please clarify in methods

We appreciate your insightful comments. The degree of resection (DOR) was based on preoperative contrast-enhanced T1-weighted and T2-weighted magnetic resonance (MR) images, as well as the first postoperative contrast-enhanced T1-weighted and T2-weighted MR images. The pre- and postoperative MR images were interpreted by a radiology specialist to assess the DOR of the tumor.1,2,3 When clinical correlation was necessary, surgical findings were referenced to evaluate the DOR.4,5 Based on this, the Materials and Methods section titled "Patients" (page 7, line 120) has been revised accordingly. 

1Multimodal Intraoperative Neurophysiological Monitoring in Intramedullary Spinal Cord Tumors: A 10-Year Single Center Experience. Cancers. 2024; 16(1):111. https://doi.org/10.3390/cancers16010111

2Surgical results of intramedullary spinal cord tumor with spinal cord monitoring to guide extent of resection. J Neurosurg Spine. 2009;10(5):404-413. doi:10.3171/2009.2.SPINE08698

3Intramedullary spinal cord ependymoma and astrocytoma: intraoperative frozen-section diagnosis, extent of resection, and outcomes. Journal of Neurosurgery: Spine SPI. 2019;30(1):133-139. doi:10.3171/2018.7.SPINE18230

4Quantitative Analysis of the Effect of Stereotactic Radiosurgery for Postoperative Residual Cervical Dumbbell Tumors: A Multicenter Retrospective Cohort Study.Neurospine. 2024;21(1):293-302. DOI: https://doi.org/10.14245/ns.2347070.535

5Intramedullary Schwannoma of the Spinal Cord: A Nationwide Analysis by the Neurospinal Society of Japan

Neurospine. 2023;20(3):747-755.DOI: https://doi.org/10.14245/ns.2346376.188

Materials and Methods

Patients

The DOR was based on preoperative contrast-enhanced T1-weighted and T2-weighted magnetic resonance (MR) images, as well as the first postoperative contrast-enhanced T1-weighted and T2-weighted MR images.(15-17) The pre- and postoperative MR images were interpreted by a radiology specialist to assess the DOR of the tumor. In cases where clinical correlation is necessary in addition to MR imaging for the assessment of DOR, surgical findings were consulted to evaluate the DOR.(18,19) Surgical records were reviewed to collect information on the timing of IOUS use and the advantages gained from its use.

Reference

15. Hongo H, Takai K, Komori T, Taniguchi M. Intramedullary spinal cord ependymoma and astrocytoma: intraoperative frozen-section diagnosis, extent of resection, and outcomes. Journal of Neurosurgery: Spine SPI. 2019;30(1):133-9.

16. Matsuyama Y, Sakai Y, Katayama Y, Imagama S, Ito Z, Wakao N, et al. Surgical results of intramedullary spinal cord tumor with spinal cord monitoring to guide extent of resection. J Neurosurg Spine. 2009;10(5):404-13.

17. Tropeano MP, Rossini Z, Franzini A, Capo G, Olei S, De Robertis M, et al. Multimodal Intraoperative Neurophysiological Monitoring in Intramedullary Spinal Cord Tumors: A 10-Year Single Center Experience. Cancers. 2024;16(1):111.

18. Hara T, Mizuno M, Hida K, Sasamori T, Miyoshi Y, Uchikado H, et al. Intramedullary Schwannoma of the Spinal Cord: A Nationwide Analysis by the Neurospinal Society of Japan. Neurospine. 2023;20(3):747-55.

19. Lee SH, Jang SW, Shin HK, Kim JH, Park D, Ha C-M, et al. Quantitative Analysis of the Effect of Stereotactic Radiosurgery for Postoperative Residual Cervical Dumbbell Tumors: A Multicenter Retrospective Cohort Study. Neurospine. 2024;21(1):293-302.

4. In case 3, why do the authors think that the US had more utility in detecting “a tiny IM tumor that remained undetected in preoperative MR”?

Thank you for your careful review. Due to the 3mm thickness of the preoperative MR images, there is a possibility that tiny tumor was not adequately covered in the MR images. Additionally, in the case of tiny tumor, there is a possibility that it was not clearly detected on contrast-enhanced T1-weighted images due to the presence of blood vessels around the spinal cord. Case 3 was a patient with von Hippel-Lindau disease who underwent surgery for a single nodule at the C7 level, as identified on preoperative MR images. However, intraoperative ultrasonography (IOUS) provided real-time imaging, which allowed for the continuous acquisition of images depending on the probe's orientation and position. This capability proved beneficial in detecting tiny nodules that were not visible on the preoperative MR images. The results of this study do not definitively establish that IOUS is superior to MR imaging in detecting tiny nodules. However, they do suggest that the use of IOUS can be beneficial in identifying tiny tumor that was not detected on preoperative MR images. This finding provides a valuable lesson regarding the potential advantages of incorporating IOUS into surgical procedures for enhanced tumor visualization.

5. Do the authors have cases where IOUS was not used? It would be very helpful to the reader to have a control group to compare degree of resection, outcome scales, etc. Otherwise the utility of the US intraoperatively is limited to the exact localization of the tumor before opening the dura.

We appreciate your insightful comments. We completely agree that establishing a control group of spinal cord tumor patients who did not use intraoperative ultrasonography (IOUS) could be beneficial for further elucidating the utility of IOUS. By comparing the extent of resection and clinical/oncologic outcomes between the groups, it would provide more substantial evidence regarding the advantages of using IOUS during surgical procedures. 

Since November 2021, our institution has been utilizing IOUS in the surgical treatment of spinal cord tumors. Consequently, the number of patients included in this study is small, and specifically regarding the detailed usage of IOUS, only three patients (7%) were residual or hidden tumors identified using IOUS, which aided in achieving total tumor resection. Therefore, it is limited in determining whether the use of IOUS significantly impacted the degree of resection. Additionally, a sufficient follow-up period has not been secured to compare clinical/oncologic outcomes effectively. Due to these reasons, a direct comparison between spinal cord tumor patients who underwent surgery with the use of IOUS and those who did not was not feasible. Our institution continues to use IOUS in spinal cord tumor surgeries beyond the period analyzed in this study. Once enough time has passed and additional data on the use of IOUS and follow-up information on the clinical and oncologic outcomes of our patients has been collected, we plan to conduct a comparative study between the group that used IOUS and the group that did not. This future research aims to further elucidate the benefits and efficacy of IOUS in spinal cord tumor surgeries. 

During the follow-up period for the patients included in this study, the clinical and oncologic outcomes were added as Table 2(page 12, line 183) and described in the Results section titled "Clinical and oncologic outcomes" (page 11, line 171). Additionally, based on these limitations, the Discussion section titled "Limitation" (page 23, line 403) has been revised accordingly.

Results

Clinical and oncologic outcomes

The mean operation time was 237.2 minutes, and gross total resection (GTR) was achieved in 40 patients (93.0%). In CA patients, motor deterioration after surgical resection was observed in 3 patients (30% of 10 patients), and CSF leakage occurred in 1 patient (10% of 10 patients). In EPN patients, postoperative motor deterioration was confirmed in 1 patient (14.3% of 7 patients), and in subependymoma (SUBEP) patient, it was confirmed in 1 patient (100%). The mean length of hospital stay (LOS) for the patients was 6.1 days, and the mean follow-up duration was 5.6 months post-surgery. At the last follow-up imaging, suspicious recurrence was detected in one CA patient (10% of

---

## [Editor Report · Decision Letter 1]

5 Jun 2024

The utility of intraoperative ultrasonography for spinal cord surgery

PONE-D-24-13339R1

Dear Dr. Kim,

We’re pleased to inform you that your manuscript has been judged scientifically suitable for publication and will be formally accepted for publication once it meets all outstanding technical requirements.

Kind regards,

Barry Kweh

Academic Editor

PLOS ONE

Additional Editor Comments (optional):

The authors have satisfactorily revised their paper and provided an adequate literature review to support their original findings.
---

## [Editor Report · Acceptance letter]

1 Jul 2024

PONE-D-24-13339R1 

PLOS ONE

Dear Dr. Kim, 

I'm pleased to inform you that your manuscript has been deemed suitable for publication in PLOS ONE. Congratulations! Your manuscript is now being handed over to our production team.

Kind regards, 

on behalf of

Dr. Barry Kweh 

Academic Editor

PLOS ONE